# INFORMATION ASYMMETRY IN KL-REGULARIZED RL

**Alexandre Galashov, Siddhant M. Jayakumar, Leonard Hasenclever, Dhruva Tirumala,
Jonathan Schwarz, Guillaume Desjardins, Wojciech M. Czarnecki, Yee Whye Teh,
Razvan Pascanu, Nicolas Heess**
DeepMind
London, UK
`{agalashov,sidmj,leonardh,dhruvat,schwarzjn,gdesjardins,`
`lejlot,ywteh,razp,heess}@google.com`

## ABSTRACT

Many real world tasks exhibit rich structure that is repeated across different parts of the state space or in time. In this work we study the possibility of leveraging such repeated structure to speed up and regularize learning. We start from the KL regularized expected reward objective which introduces an additional component, a default policy. Instead of relying on a fixed default policy, we learn it from data. But crucially, we restrict the amount of information the default policy receives, forcing it to learn reusable behaviours that help the policy learn faster. We formalize this strategy and discuss connections to information bottleneck approaches and to the variational EM algorithm. We present empirical results in both discrete and continuous action domains and demonstrate that, for certain tasks, learning a default policy alongside the policy can significantly speed up and improve learning.

## 1 INTRODUCTION

For many interesting reinforcement learning tasks, good policies exhibit similar behaviors in different contexts, behaviors that need to be modified only slightly or occasionally to account for the specific task at hand or to respond to information becoming available. For example, a simulated humanoid in navigational tasks is usually required to walk – independently of the specific goal it is aiming for. Similarly, an agent in a simulated maze tends to primarily move forward with occasional left/right turns at intersections.

This intuition has been explored across multiple fields, from cognitive science (e.g. Kool & Botvinick, 2018) to neuroscience and machine learning. For instance, the idea of bounded rationality (e.g. Simon, 1956) emphasizes the cost of information processing and the presence of internal computational constraints. This implies that the behavior of an agent minimizes the need to process information, and more generally trades off task reward with computational effort, resulting in structured repetitive patterns. Computationally, these ideas can be modeled using tools from information and probability theory (e.g. Tishby & Polani, 2011; Ortega & Braun, 2011; Still & Precup, 2012; Rubin et al., 2012; Ortega & Braun, 2013; Tiomkin & Tishby, 2017), for instance, via constraints on the channel capacity between past states and future actions in a Markov decision process.

In this paper we explore this idea, starting from the KL regularized expected reward objective (e.g. Todorov, 2007; Toussaint, 2009; Kappen et al., 2012; Rawlik et al., 2012; Levine & Koltun, 2013; Teh et al., 2017), which encourages an agent to trade off expected reward against deviations from a prior or default distribution over trajectories. We explore how this can be used to inject subjective knowledge into the learning problem by using an informative default policy that is learned alongside the agent policy This default policy encodes default behaviours that should be executed in multiple contexts in absence of addi-

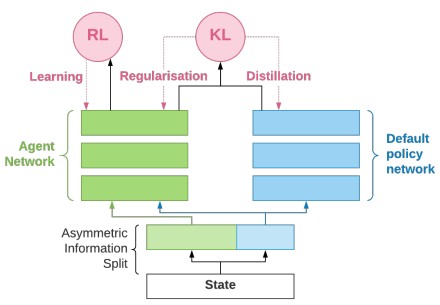

Figure 1: Default policy-agent architecture.

tional task information and the objective forces the
learned policy to be structured in alignment with the default policy.

To render this approach effective, we introduce an information asymmetry between the default and
agent policies, preventing the default policy from accessing certain information in the state. This
prevents the default policy from collapsing to the agent's policy. Instead, the default policy is forced
to generalize across a subset of states, implementing a form of default behavior that is valid in the
absence of the missing information, and thereby exerting pressure that encourages sharing of behavior
across different parts of the state space.

Figure 1 illustrates the proposed setup, with asymmetry imposed by hiding parts of the state from
the default policy. We investigate the proposed approach empirically on a variety of challenging
problems including both continuous action problems such as controlling simulated high-dimensional
physical embodied agents, as well as discrete action visual mazes. We find that even when the agent
and default policies are learned at the same time, significant speed-ups can be achieved on a range
of tasks. We consider several variations of the formulation, and discuss its connection to several
ideas in the wider literature, including information bottleneck, and variational formulations of the
EM algorithm for learning generative models.

## 2 KL AND ENTROPY REGULARIZED REINFORCEMENT LEARNING

Throughout this paper we use $s_t$ and $a_t$ to denote the state and action at time step $t$, and $r(s, a)$ the
instantaneous reward for the agent if it executes action $a$ in state $s$. We denote the history up to time $t$
by $x_t = (s_1, a_1, \ldots, s_t)$, and the whole trajectory by $\tau = (s_1, a_1, s_2, \ldots)$. Our starting point is the
KL regularized expected reward objective

$$\mathcal{L}(\pi, \pi^0) = \mathbb{E}_{\pi_\tau} \left[ \sum_t \gamma^t r(s_t, a_t) - \alpha \gamma^t \mathsf{KL} \left[ \pi(a_t|x_t) \| \pi^0(a_t|x_t) \right] \right], \quad (1)$$

where $\pi$ is the agent policy (parameterized by $\theta$ and to be learned), $\pi^0$ the default policy, and $\mathbb{E}_{\pi_\tau}[\cdot]$
is taken with respect to the distribution $\pi_\tau$ over trajectories defined by the agent policy and system
dynamics: $\pi_\tau(\tau) = p(s_1) \prod_t \pi(a_t|x_t) p(s_{t+1}|s_t, a_t)$. Note that our policies are history-dependent.
$\mathsf{KL}[\pi(a_t|x_t) \| \pi^0(a_t|x_t)]$ is the Kullback-Leibler (KL) divergence between the agent policy $\pi$ and a
default or prior policy $\pi^0$ given history $x_t$. The discount factor is $\gamma \in [0, 1]$ and $\alpha$ is a hyperparameter
scaling the relative contributions of both terms.

Intuitively, this objective expresses the desire to maximize the reward while also staying close to a
reference behaviour defined by $\pi^0$. As discussed later, besides being a convenient way to express
a regularized RL problem, it also has deep connections to probabilistic inference. One particular
instantiation of eq. (1) is when $\pi^0$ is the uniform distribution (assuming a compact action space). In
this case one recovers, up to a constant, the entropy regularized objective (e.g. Ziebart, 2010; Fox
et al., 2015; Haarnoja et al., 2017; Schulman et al., 2017a; Hausman et al., 2018):

$$\mathcal{L}_H(\pi) = \mathbb{E}_{\pi_\tau} \left[ \sum_t \gamma^t r(s_t, a_t) + \alpha \gamma^t \mathsf{H}[\pi(a_t|x_t)] \right]. \quad (2)$$

This objective has been motivated in various ways: it prevents the policy from collapsing to a
deterministic solution thus improving exploration, it encourages learning of multiple solutions to a
task which can facilitate transfer, and it provides robustness to perturbations and model mismatch.
One approximation of the entropy regularized objective is for the history dependent entropy to be
used as an additional (auxiliary) loss to the RL loss; this approach is widely used in the literature
(e.g. Williams & Peng, 1991; Mnih et al., 2016). While the motivations for considering the entropy
regularized objective are intuitive and reasonable, the choice of regularizing towards an uniform
policy is less obvious, particularly in cases with large or high dimensional action spaces. In this work
we explore whether regularization towards more sophisticated default policies can be advantageous.

Both objectives (1) and (2) can be generalized beyond the typical Markov assumption in MDPs.
In particular, additional correlations among actions can be introduced, e.g. using latent variables
Hausman et al. (2018). This can be useful when, as discussed below, either $\pi^0$ or $\pi$ are not given full
access to the state, rendering the setup partially observed. In the following we will not explore such
extensions, though note that we do work with policies $\pi(a_t|x_t)$ and $\pi^0(a_t|x_t)$ that depend on history
$x_t$.

## 3 LEARNING DEFAULT POLICIES

Many works that consider the KL regularized objective either employ a simple or fixed default policy or directly work with the entropy formulation (e.g. Rubin et al., 2012; Fox et al., 2015; Haarnoja et al., 2017; Hausman et al., 2018). In contrast, here we will be studying the possibility of learning the default policy itself, and the form of the subjective knowledge that this introduces to the learning system. Our guiding intuition, as described earlier, is the notion of a default behaviour that is executed in the absence of additional goal-directed information. Instances which we explore in this paper include a locomotive body navigating to a goal location where the locomotion pattern depends largely on the body configuration and less so on the goal, and a 3D visual maze environment with discrete actions, where the typical action includes forward motion, regardless of the specific task at hand.

To express the notion of a default behavior, which we also refer to as "goal-agnostic" (although the term should be understood very broadly), we consider the case where the default policy $\pi^0$ is a function (parameterized by $\phi$) of a subset of the interaction history up to time $t$, i.e. $\pi^0(a_t|x_t) = \pi^0(a_t|x_t^{\mathcal{D}})$, where $x_t^{\mathcal{D}}$ is a subset of the full history $x_t$ and is the goal-agnostic information that we allow the default policy to depend on. We denote by $x_t^{\mathcal{G}}$ the other (goal-directed) information in $x_t$ and assume that the full history is the disjoint union of both. The objective (1) specializes to:

$$\mathcal{L}(\pi, \pi^0) = \mathbb{E}_{\pi_\tau} \left[ \sum_t \gamma^t r(s_t, a_t) - \alpha \gamma^t \mathsf{KL} \left[ \pi(a_t|x_t) \| \pi^0(a_t|x_t^{\mathcal{D}}) \right] \right], \tag{3}$$

To give a few examples: If $x_t^{\mathcal{D}}$ is empty then the default policy does not depend on the history at all (e.g. uniform policy). If $x_t^{\mathcal{D}} = a_{1:t-1}$ then it depends only on past actions. In multitask learning $x_t^{\mathcal{G}}$ can be the task identifier, while $x_t^{\mathcal{D}}$ the state history. And finally, in continuous control $x_t^{\mathcal{D}}$ can contain proprioceptive information about the body, while $x_t^{\mathcal{G}}$ contains exteroceptive (goal-directed) information (e.g. vision).

By hiding information from the default policy, the system forces the default policy to learn the average behaviour over histories $x_t$ with the same value of $x_t^{\mathcal{D}}$. If $x_t^{\mathcal{D}}$ hides goal-directed information, the default policy will learn behaviour that is generally useful regardless of the current goal. We can make this precise by noting that optimizing the objective (1) with respect to $\pi^0$ amounts to supervised learning of $\pi^0$ on trajectories generated by $\pi_\tau$, i.e. this is a distillation process from $\pi_\tau$ to $\pi^0$ (Hinton et al., 2015; Rusu et al., 2016; Parisotto et al., 2016; Teh et al., 2017). In the nonparametric case, the optimal default policy $\pi_*^0$ can be derived as:

$$\pi_*^0(a_t|x_t^{\mathcal{D}}) = \frac{\sum_{\tilde{t}} \gamma^{\tilde{t}} \int \left( \mathbb{1}(x_t^{\mathcal{D}} = \tilde{x}_{\tilde{t}}^{\mathcal{D}}) \pi(a_t|\tilde{x}_{\tilde{t}}) \right) \pi_\tau(\tilde{x}_{\tilde{t}}) d\tilde{x}_{\tilde{t}}}{\sum_{\tilde{t}} \gamma^{\tilde{t}} \int \left( \mathbb{1}(x_t^{\mathcal{D}} = \tilde{x}_{\tilde{t}}^{\mathcal{D}}) \right) \pi_\tau(\tilde{x}_{\tilde{t}}) d\tilde{x}_{\tilde{t}}}, \tag{4}$$

where $\pi_\tau(\tilde{x}_{\tilde{t}})$ is the probability of seeing history $\tilde{x}_{\tilde{t}}$ at time step $\tilde{t}$ under the policy $\pi$, and the indicator $\mathbb{1}(x_t^{\mathcal{D}} = \tilde{x}_{\tilde{t}}^{\mathcal{D}})$ is 1 if the goal-agnostic information of the two histories matches and 0 otherwise.

It is also worth considering the effect of the objective eq. (3) on the learned policy $\pi$. Since $\pi^0$ is learned alongside $\pi$ and not specified in advance, this objective does not favor any particular behavior a priori. Instead it will encourage a solution in which similar behavior will be executed in different parts of the state space that are similar as determined by $x_t^{\mathcal{D}}$, since the policy $\pi$ is regularized towards the default policy $\pi^0$. More generally, during optimization of $\pi$ the default policy effectively acts like a shaping reward while the entropy contained in the KL discourages deterministic solutions.

## 4 CONNECTION TO INFORMATION BOTTLENECK AND VARIATIONAL EM

### 4.1 INFORMATION BOTTLENECK

Reinforcement learning objectives with information theoretic constraints have been considered by multiple authors (Tishby & Polani, 2011; Still & Precup, 2012; Tiomkin & Tishby, 2017). Such constraints can be motivated by the internal computational limitations of the agent, which limit the rate with which information can be extracted from states (or observations) and translated into actions. Such capacity constraints can be expressed via an information theoretic regularization term that is added to the expected reward. Specializing to our scenario, where the "information flow" to be controlled is between the goal-directed history information $x_t^G$ and action $a_t$ (so that the agent prefers

default, goal-agnostic, behaviour), consider the objective:

$$\mathcal{L}_I = \mathbb{E}_{\pi_\tau} \left[ \sum_t \gamma^t r(s_t, a_t) - \alpha \gamma^t \mathsf{MI}[x_t^{\mathcal{G}}, a_t | x_t^{\mathcal{D}}] \right], \tag{5}$$

where $\mathsf{MI}[x_t^{\mathcal{G}}, a_t | x_t^{\mathcal{D}}]$ is the conditional mutual information between $x_t^{\mathcal{G}}$ and $a_t$ given $x_t^{\mathcal{D}}$. The conditional mutual information can be upper bounded:

$$\mathbb{E}_{\pi_\tau}[\mathsf{MI}[x_t^{\mathcal{G}}, a_t | x_t^{\mathcal{D}}]] = \mathbb{E}_{\pi_\tau} \left[ \log \frac{\pi_\tau(x_t^{\mathcal{G}} | x_t^{\mathcal{D}}) \pi(a_t | x_t^{\mathcal{G}}, x_t^{\mathcal{D}})}{\pi_\tau(x_t^{\mathcal{G}} | x_t^{\mathcal{D}}) \pi_\tau(a_t | x_t^{\mathcal{D}})} \right] \leqslant \mathbb{E}_{\pi_\tau} \left[ \log \frac{\pi(a_t | x_t^{\mathcal{G}}, x_t^{\mathcal{D}})}{\pi^0(a_t | x_t^{\mathcal{D}})} \right] \tag{6}$$

where the inequality is from the fact that the KL divergence $\mathsf{KL}[\pi_\tau(a_t | x_t^{\mathcal{D}}) \| \pi^0(a_t | x_t^{\mathcal{D}})]$ is positive (see Alemi et al., 2016). Re-introducing this into (5) we find that the KL regularized objective in eq. (3) can be seen as a lower bound to eq. (5), where the agent has a capacity constraint on the channel between goal-directed history information and (future) actions. See section A in the appendix for a generalization including latent variables. In this light, we can see our work as a particular implementation of the information bottleneck principle, where we penalize the dependence on the information that is hidden from the default policy.

## 4.2 VARIATIONAL EM

The above setup also bears significant similarity to the training of variational autoencoders (Kingma & Welling, 2013; Rezende et al., 2014) and, more generally the variational EM framework for learning latent variable models (Dempster et al., 1977; Neal & Hinton, 1999). The setup is as follows. Given observations $\mathcal{X} = \{x_1, \ldots x_N\}$ the goal is to maximize the log marginal likelihood $\log p_\theta(\mathcal{X}) = \sum_i \log p_\theta(x_i)$ where $p_\theta(x) = \int p_\theta(x, z) dz$. This marginal likelihood can be bounded from below by $\sum_i \mathbb{E}_{q_\phi(z|x_i)}[\log p_\theta(x_i | z) - \log \frac{q_\phi(z|x_i)}{p_\theta(z)}]$ with $q_\phi(z|x_i)$ being a learned approximation to the true posterior $p_\theta(z|x_i)$. This lower bound exhibits a similar information asymmetry between $q$ and $p$ as the one introduced between $\pi$ and $\pi^0$ in the objective in eq. (3). In particular, in the multi-task case discussed in section 3 with one task per episode, $x_i$ can be seen to take the role of the task, $\log p(x_i | z)$ that of the task reward, $q(z|x_i)$ that of task conditional policy, and $p(z)$ the default policy. Therefore maximizing eq. (3) can then be thought of as learning a generative model of behaviors that can explain the solution to different tasks.

## 5 ALGORITHM

In practice the objective in eq. 3 can be optimized in different ways. A simple approach is to perform alternating gradient ascent in $\pi^0$ and $\pi$. Optimizing $\mathcal{L}$ with respect to $\pi^0$ amounts to supervised learning with $\pi$ as the data distribution (distilling $\pi$ into $\pi^0$). Optimizing $\pi$ given $\pi^0$ requires solving a regularized expected reward problem which can be achieved with a variety of algorithms (Schulman et al., 2017a; Teh et al., 2017; Haarnoja et al., 2017; Hausman et al., 2018; Haarnoja et al., 2018).

The specific algorithm choice in our experiments depends on the type of environment. For the continuous control domains we use SVG(0) (Heess et al., 2015) with experience replay and a modification for the KL regularized setting (Hausman et al., 2018; Haarnoja et al., 2018). The SVG(0) algorithm learns stochastic policies by backpropagation from the action-value function. We estimate the action value function using $K$-step returns and the Retrace operator for low-variance off-policy correction (see Munos et al. (2016); as well as Hausman et al. (2018); Riedmiller et al. (2018b)). For discrete action spaces we use a batched actor-critic algorithm (see Espeholt et al. (2018)). The algorithm employs a learned state-value function and obtains value estimates for updating the value function and advantages for computing the policy gradient using $K$-step returns in combination with the V-trace operator for off-policy correction. All algorithms are implemented in batched distributed fashion with a single learner and multiple actors. In algorithm 1 we provide pseudo-code for actor-critic version of the algorithm with $K$-step returns. Details of the off-policy versions of the algorithms for continuous and discrete action spaces can be found in the appendix (section D).

## 6 RELATED WORK

There are several well established connections between certain formulations of the reinforcement learning literature and concepts from the probabilistic modeling literature. The formalisms are often closely related although derived from different intuitions, and with different intentions.

---

**Algorithm 1** Simple actor-critic algorithm with K-step returns

---

policy: $\pi_\theta$, initial parameters $\theta^0$
default policy: $\pi_\phi^0$; initial parameters $\phi^0$
Q-function: $Q_\psi$; initial parameters $\psi^0$
**for** j=1, ... **do**
    **for** t = 0, K, 2K, ... T **do**
        rollout partial trajectory: $\tau_{t:t+K} = (s_t, a_t, r_t \ldots r_{t+K})$
        compute KL: $\widehat{\mathsf{KL}}_{t'} = \mathsf{KL}[\pi(\cdot|s_{t'})\|\pi^0(\cdot|s_{t'})]$
        Estimate boostrap value: $\hat{V} = \mathbb{E}_{\pi(\cdot|s_{t+K})}[Q(s_{t+K}, a)] - \alpha\widehat{\mathsf{KL}}_{t+K}$
        Estimate Q targets: $\hat{Q}_{t'} = \sum_{t''=t'}^{t+K-1}(r_{t''} - \alpha\widehat{\mathsf{KL}}_{t''}) + \hat{V}$
        Agent policy loss: $\hat{L}_\pi = \sum_{t'=t}^{t+K-1}\mathbb{E}_{\pi(\cdot|s_{t'})}[Q(s_{t'}, a)] - \alpha\widehat{\mathsf{KL}}_{t'}$
        Q-value loss: $\hat{L}_Q = \sum_{t'=t}^{t+K-1}\|\hat{Q}_{t'} - Q(s_{t'}, a_{t'})\|^2$
        Default policy loss: $\hat{L}_{\pi^0} = \sum_{t'=t}^{t+K-1}\widehat{\mathsf{KL}}_{t'}$
        $\theta \leftarrow \theta + \beta_\pi\nabla_\theta\hat{L}_\pi \qquad \phi \leftarrow \phi - \beta_{\pi^0}\nabla_\phi\hat{L}_{\pi^0} \qquad \psi \leftarrow \psi - \beta_Q\nabla_\psi\hat{L}_Q$
    **end for**
**end for**

---

*Maximum entropy reinforcement learning*, stochastic optimal control, and related approaches build on the observation that some formulation of the reinforcement learning problem can be interpreted as exact or approximate variational inference in a probabilistic graphical model in which the reward function takes the role of log-likelihood (e.g. Ziebart, 2010; Kappen et al., 2012; Toussaint, 2009). While the exact formulation and algorithms vary, they result in an entropy or KL regularized expected reward objective. These algorithms were originally situated primarily in the robotics and control literature but there has been a recent surge in interest in deep reinforcement learning community (e.g. Fox et al., 2015; Schulman et al., 2017a; Nachum et al., 2017a; Haarnoja et al., 2017; Hausman et al., 2018; Haarnoja et al., 2018).

Related but often seen as distinct is the familiy of *expectation maximization policy search algorithms* (e.g. Peters et al., 2010; Rawlik et al., 2012; Levine & Koltun, 2013; Montgomery & Levine, 2016; Chebotar et al., 2016; Abdolmaleki et al., 2018). These cast policy search as an alternating optimization problem similar to the EM algorithm for learning probabilistic models. They differ in the specific implementation of the equivalents of the E and M steps; intuitively the default policy is repeatedly replaced by a new version of the policy.

The DISTRAL algorithm (Teh et al., 2017) as well as the present paper can be seen as taking an intermediate position: unlike in the class of RL-as-inference algorithms the default policy is not fixed but learned, but unlike in the classical EM policy search the final result of the optimization remains regularized since the default policy is constrained relative to the policy. As explained above this can be seen as analogous to the relative roles of learned model and observation specific posterior in fitting a generative model. Similar to DISTRAL, Divide and Conquer (Ghosh et al., 2018) learns an ensemble of policies, each specializing to a particular context, which are regularized towards one another via a symmetric KL penalty, with the behavior of the ensemble distilled to a single fixed policy. In concurrent work Goyal et al. (2019) propose an information bottleneck architecture for policies with latent variables that leads to a KL-regularized formulation similar to the one described in Appendix A.2. The information bottleneck is implemented in latent space and the default policy is obtained by marginalization with a goal-agnostic prior.

An important feature of EM policy search and other policy gradient algorithms is the presence of a KL constraint that limits the relative change of the policy to some older version across iterations to control for the rate of change in the policy (e.g. Schulman et al., 2015; Heess et al., 2015; Schulman et al., 2017b; Heess et al., 2017; Nachum et al., 2017b). The constraint can be implemented in different ways, and collectively the algorithms are often classified as "trust region" methods. Note that for a KL regularized objective to be a trust region (Nocedal & Wright, 2006), additional assumptions need to hold. In principle, as an optimization technique, the critical points of the KL regularized objective for some function $f(\theta)$ have to be, provably, the same as for the non-regularized objective. This is not trivial to show unless the trust region for step $k$ is around $\theta_k$. In our case, there is no such

guarantee even if we remove the asymmetry in information between default policy and policy or make the default policy be an old copy of the policy.

Other related works motivated from an optimization perspective include Deep Mutual Learning (Zhang et al., 2018) applied in supervised learning, where KL-regularization is used with a learned prior that receives the same amount of information as the trained model. Kirkpatrick et al. (2017) introduces EWC to address catastrophic forgetting, where a second order Taylor expansion of the KL, in a KL-regularized objective, forces the main policy to stay close to solutions of previously encountered tasks. Czarnecki et al. (2018) also relies on a KL-regularized objective to ensure policies explored in a curriculum stay close to each other.

Conceptually distinct but formally closely related to maximum entropy and KL-regularized formulations are computational models of bounded rationality (e.g. Tishby & Polani, 2011; Ortega & Braun, 2011; Still & Precup, 2012; Rubin et al., 2012; Tiomkin & Tishby, 2017) which introduce information constraints to account for the agent's internal computational constraints on its ability to process information. As discussed in section 4 the present formulation can be seen as a more general formulation of the idea.

# 7 CONTINUOUS CONTROL EXPERIMENTS

In our experiments, we study the effect of using a learned default policy to regularize the behavior of our agents, across a wide range of environments spanning sparse and dense reward tasks. In particular, we evaluate the impact of conditioning the default policy on various information sets $x^{\mathcal{D}}$ on the learning dynamics, and evaluate the potential of pretrained default policies for transfer learning. In these experiments, we consider two streams of information which are fed to our agents: **task specific information** (task) and **proprioception** (proprio), corresponding to **walker** (body) specific observations (joint angles etc.).

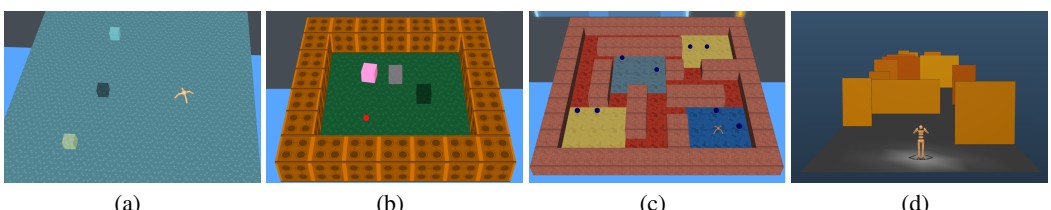

| (a) | (b) | (c) | (d) |

Figure 2: **Tasks visualization**. (a): Go to one of K target tasks, with *quadruped*; (b): Move one box to one of K targets task, with *jumping ball* (red); (c): Foraging in the maze task, with *quadruped*; (d): Walls task with *humanoid*, where the goal is avoid walls while running through a terrain.

We consider three walkers: *jumping ball* with 3 degrees of freedom (DoF) and 3 actuators; *quadruped* with 12 DoF and 8 actuators; *humanoid* with 28 DoF and 21 actuators. The task is specified to the agent either via an additional feature vector (referred to as *feature-tasks*) or in the form of visual input (*vision-task*). The tasks differ in the type of reward: in *sparse* reward tasks a non-zero reward is only given when a (sub-)goal is achieved (e.g. the target was reached); in *dense* reward tasks smoothly varying shaping reward is provided (e.g. negative distance to the target). We consider the following tasks.

**Walking task**, a *dense-reward* task based on *features*. The walker needs to move in one of four randomly sampled directions, with a fixed speed; the direction being resampled half-way through the episode. **Walls task**, a *dense-reward vision-task*. Here the walker has to traverse a corridor while avoiding walls. **Go to one of K targets task**, a *sparse-reward feature-based* task. The walker has to go to one of K randomly sampled targets. For K=1, the target can either reappear within the episode (referred to as the *moving target* task) or the episode can end upon reaching the target. **Move one box to one of K targets**, a *sparse-reward feature-based-task*. The walker has to move a box to one of K targets, and optionally, go on to one of the remaining targets. The latter is referred to as the *move one box to one of K targets and go to another target*). **Foraging in the maze task**, a *sparse-reward*

*vision-task*. The walker collects apples in a maze. Figure 2 shows visualizations of the walkers and some of the tasks. Refer to appendix C for more details.

**Experimental Setup**    As baseline, we consider policies trained with standard entropy regularization. When considering the full training objective of eq. 1, the default policy network shares the same structure as the agent's policy. In both cases, hyper-parameters are optimized on a per-task basis. We employ a distributed actor-learner architecture (Espeholt et al., 2018): actors execute recent copies of the policy and send data to a replay buffer of fixed size; while the learner samples short trajectory windows from the replay and computes updates to the policy, value, and default policy. We experimented with a number of actors in $\{32, 64, 128, 256\}$ (depending on the task) and a single learner. Results with a single actor are presented in appendix B. Unless otherwise mentioned, we plot average episodic return as a function of the number of environment transitions processed by the learner[1]. Each experiment is run with five random seeds. For more details, see appendix D.2

We consider three information sets passed to the default policy: **proprioceptive**, receiving only proprioceptive information; **task-subset**, receiving proprioceptive and a subset of task-specific information; **full-information**, receiving the same information as the policy.

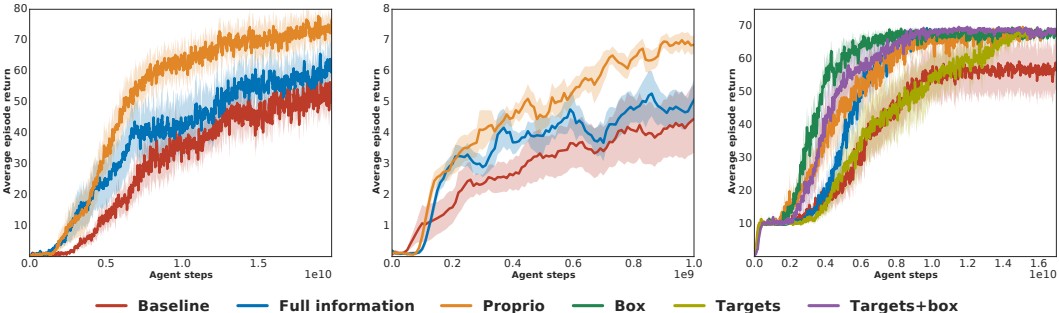

Figure 3: **Results for the *sparse-reward* tasks with *complex walkers*. Left**: go to moving target task with *humanoid*. **Center**: foraging in the maze results with *quadruped*. **Right**: moving one box to one of two targets and go to another target task with *quadruped*. The legends denote additional to the proprioception, information passed to the default policy (except baseline, where we do not use default policy).

The main finding of our experiments is that the default policy with limited task information provides considerable speed-up in terms of learner steps for the *sparse-reward* tasks with *complex walkers* (*quadruped*, *humanoid*). The results on these tasks are presented in figure 3. More cases are covered in the appendix E.

Overall, the proprioceptive default policy is very effective and gives the biggest gains in the majority of tasks. Providing additional information to the default policy, leads to an improvement only in a small number of cases (figure 3, right and appendix E.3). In these cases, the additional information (e.g. box position), adds useful inductive bias for the policy learning. For the *dense-reward* tasks or for a simple walker body adding the default policy has limited or no effect (see appendix E.1, E.2). We hypothesize that the absence of gain is due to the relative simplicity of the regular policy learning versus the KL-regularized setup. In the case of *dense-reward* tasks the agent has a strong reward signal. For simple walkers, the action space is too simple to require sophisticated exploration provided by the default policy. Finally, with full information in the default policy, the optimal default policy would exactly copy the agent policy, which would not provide additional learning signal beyond the regular policy learning. In all these cases, the default policy will not be forced to generalize across different contexts and hence not provide a meaningful regularization signal.

We analyze the agent behavior on the go to moving target task with a *quadruped* walker. We illustrate the agent trajectory for this task in figure 4, left. The red dot corresponds to the agent starting position. The green stars on the left and central figures correspond to the locations of the targets with

---

[1]Note that due to the distributed setup with experience replay this number does not directly translate to the number of environment steps executed or gradient updates performed (the latter can be computed dividing the steps processed by batch size and unroll length).

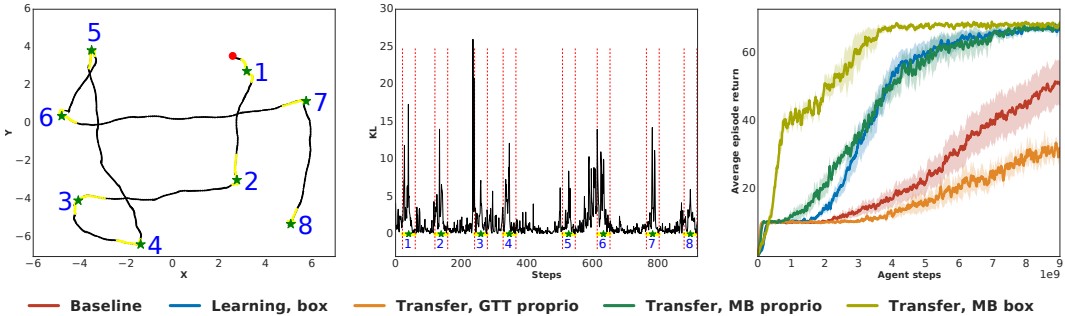

Figure 4: **Behavior analysis and transfer results**. **Left**: the trajectory of the agent on go to moving target task with *quadruped*. **Center**: KL divergence from the agent policy to the proprioceptive default policy plotted over time for the same trajectory. **Right**: Performance of the transfer on move one box to one of 3 targets task with *quadruped*. The legend whether the default policy is learned or is transferred. Furthermore, it specifies the task from which the default policy is transferred as well as additional information other than the proprioceptive information that the default policy is conditioned on, if any.

blue numbers indicating the order of achieving the targets. The yellow dots on the left and central curves indicate the segment (of 40 time steps) near the target. In figure 4, center, we show the KL divergence, $KL[\pi\|\pi^0]$, from the agent policy to the proprioceptive default policy. We observe that for the segments which are close to the target (yellow dots near green star), the value of the KL divergence is high. In these segments the walker has to stop and turn in order to go to another target. It represents a deviation from the standard, walking behavior, and we can observe it as spikes in the KL. Furthermore, for the segments between the targets, e.g. 4 –> 5, the KL is much lower.

**Default Policy Transfer** We additionally explore the possibility of reusing pretrained default policies to regularize learning on new tasks. Our transfer task is *moving one box to one of 2 targets and going to another target task* with the *quadruped*. We consider different default policies: **GTT proprio**: proprioceptive information only trained on *going to moving target task* (*GTT*); **MB proprio**: proprioceptive information only trained on *moving one box to one target task* (*MB*); **MB box**: similar *MB proprio*, but with box position information as additional input. The results are given in figure 4, right. We observe a significant improvement in learning speed transferring the pretrained default policies to the new task. Performance improves as the trajectory distribution modeled by the default policy is closer to the one appropriate for the transfer task (compare GTT proprio with MB proprio; and MB proprio with MB box).

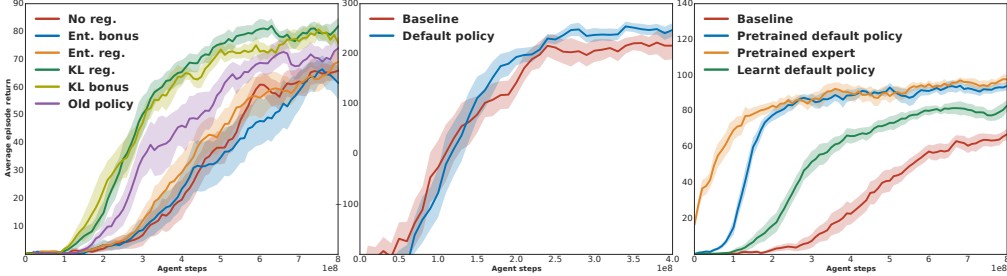

Figure 5: **Ablations**. **Left**: Comparing various regularization schemes. **Center**: Benefits of default policy vanish when using (dense) shaping rewards. **Right**: Optimistic baselines comparing pretrained default policies.

**Ablative Analysis** To gain deeper insights into our method, we compare different forms of regularization of the standard RL objective: *entropy bonus* - adding an entropy term $H(\pi(\cdot|s_t))$ to the per-timestep actor loss; *entropy regularization* - optimizing the objective (2); *KL bonus* - adding the KL-divergence term $KL[\pi(a_t|s_t)\|\pi^0(a_t|s_t)]$ from the agent policy to the default one to the

per-timestep actor loss; *KL-regularization* - optimizing the objective (1); *KL-regularization to the old policy* - optimization of the objective 1 where regularization is done wrt. an older version of the main policy (updated every 100 steps). The default policy receives only proprioceptive information in these experiments. The task is *go to moving target*. As can be seen in Figure 5 left, all three KL-based variants improve performance over the baseline, but regularizing against the information restricted default policy outperforms regularization against an old version of the policy.

Figure 5 center, demonstrates that the benefit of the default policy depends on the reward structure. When replacing the sparse reward with a dense shaping reward, proportional to the inverse distance from the walker to the target, our method and the baseline perform similarly, which is consistent with dense-reward results.

Finally, we assess the benefit of the KL-regularized objective 1 when used with an idealized default policy. We repeat the go-to-target experiment with a pretrained default policy on the same task. Figure 5 right, shows a significant difference between the baseline and different regularization variants: using the pretrained default policy, learning the default policy alongside the main policy or using a pretrained *expert* (default policy with access to the full state). This suggests that large gains may be achievable in situations when a good default policy is known *a priori*. We performed the same analysis for the dense reward but we did not notice any gain. The speed-up from regularizing to the pretrained *expert* is significant, however it corresponds to regularizing against an existing solution and can thus primarily be used as a method to speed-up the experiment cycles, as it was demonstrated in kickstarting framework (Schmitt et al., 2018).

Finally, we study impact of the direction of the KL in objective 1 on the learning dynamics. Motivated by the work in policy distillation (Rusu et al., 2016) we flip the KL and use $\mathsf{KL}\left[\pi^0(a_t|s_t)\|\pi(a_t|s_t)\right]$ instead of the described before $\mathsf{KL}\left[\pi(a_t|s_t)\|\pi^0(a_t|s_t)\right]$. The experiments showed that there was no significant difference between these regularization schemes, which suggests that the idea of learned default policy can be viewed from student-teacher perspective, where default policy plays the role of the teacher. This teacher can be used in a new task. For the details, please refer to the appendix E.6.

# 8 DISCRETE ACTION SPACES EXPERIMENTS

We also evaluate our method on the DMLab-30 set of environments. DMLab (Beattie et al., 2016) provides a suite of rich, first-person environments with tasks ranging from complex navigation and laser-tag to language-instructed goal finding. Recent works on multitask training (Espeholt et al., 2018) in this domain have used a form of batched-A2C with the V-trace algorithm to maximize an approximation of the entropy regularized objective described earlier, where the default policy is a uniform distribution over the actions.

Typically, the agent receives visual information at each step, along with an instruction channel used in a subset of tasks. The agent receives no task identifier. We adopt the architecture employed in previous work (Espeholt et al., 2018) in which frames, past actions and rewards are passed successively through a deep residual network and LSTM, finally predicting a policy and value function. All our experiments are tuned with population-based training (Jaderberg et al., 2017). Further details are provided in appendix D.1.

DMLab exposes a large action space, specifically the cartesian product of atomic actions along seven axes. However, commonly a human-engineered restricted subset of these actions is used at training and test time, simplifying the exploration problem for the agent. For example, the used action space has a *forward bias*, with more actions resulting in the agent moving forward rather than backwards. This helps with exploration in navigation tasks, where even a random walk can get the agent to move away from the starting position. The uniform default policy is used on top of this human engineered small action space, where its semantics are clear.

In this work, we instead consider a much larger combinatorial space of actions. We show that a pure uniform default policy is in fact unhelpful when human knowledge is removed from defining the right subset of actions to be uniform over, and the agent under-performs. Learning the default policy, even in the extreme case when the default policy is not conditioned on any state information, helps recovering which actions are worth exploring and leads to the emergence of a useful action space without any hand engineering.

Figure 6 shows the results of our experiments. We consider a flat action space of 648 actions, each moving the agent in different spatial dimensions. We run the agent from (Espeholt et al., 2018) as baseline which is equivalent to considering the default policy to be a uniform distribution over the 648 actions, and three variants of our approach, where the default policy is actually learnt.

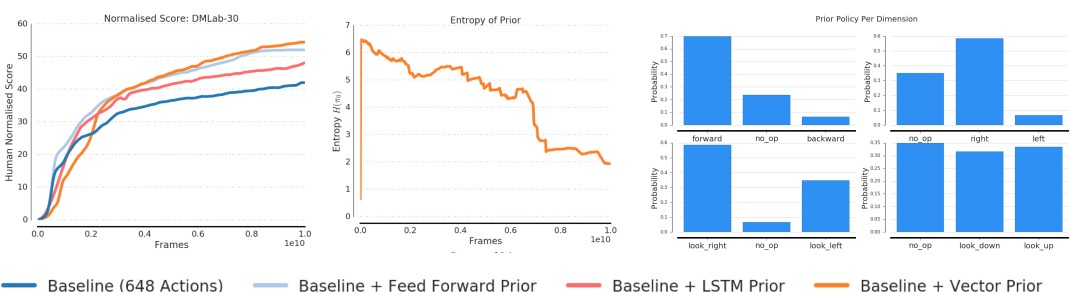

Figure 6: **DMLab30**. Left, comparison between baseline (same as Espeholt et al. (2018)) that uses uniform distribution over actions as a default policy and three different possible default policies. Center, the entropy for the vector default policy over learning. Right, marginalized distribution over few actions of interest for the vector default policy.

For **feed forward default policy**, while the agent is recurrent, the default policy is not. That is the policy $\pi$ is conditioned on the full trace of observed states $s_1, a_1, ..s_t$, while the default policy $\pi^0$ is conditioned only on the current frame $a_{t-1}, s_t$. Given that most of the 30 tasks considered require memory in order to be solvable, the default policy has to generalize over important task details. **LSTM default policy** on the other hand, while being recurrent as the agent, it observes only the previous action $a_{t-1}$ and does not receive any other state information. In this instance, the default policy can only model the most likely actions given recent behaviour $a_1, ..a_{t-1}$ in absence of any visual stimuli. For example, if previous actions are *moving forward*, the default policy might predict *moving forward* as the next action too. This is because the agent usually moves consistently in any given direction in order to navigate efficiently. Finally, the **vector default policy** refers to a default policy that is independent of actions and states (i.e. average behaviour over all possible histories of states and actions).

Using any of the default policies outperforms the baseline, with LSTM default policy slightly underperforming compared with the others. The vector default policy performs surprisingly well, highlighting that for DMLab defining a meaningful action space is extremely important for solving the task. Our approach can provide a mechanism for identifying this action space without requiring human expert knowledge on the tasks. Note in middle plot, figure 6, that the entropy of the default policy over learning frames goes down, indicating that the default policy becomes peaky and is quite different from the uniform distribution which the baseline assumes. Note that when running the same experiments with the original human-engineered smaller action space, no gains are observed. This is similar to the continuous control setup, corresponding to changing the walker to a simple one and hence converting the task into a denser reward one.

Additionally, in figure 6 right, for the vector default policy, we show the probability of a few actions of interest by marginalizing over all other actions. We notice that the agent has a tendency of moving forward 70%, while moving backwards is quite unlikely 10%. The default policy discovers one element of the human defined action space, namely *forward-bias* which is quite useful for exploring the map. The uniform bias would put same weight for moving forward as for moving backwards, making exploration harder. We also note that the agent has a tendency to turn right and look right. Given that each episode involves navigating a new sampled map, such a bias provides a meaningful exploration boost, as it suggest a *following the wall* strategy, where at any new intersection the agent always picks the same turning direction (e.g. right) to avoid moving in circles. But as expected, since neither looking up or looking down provides any advantage, these actions are equally probable.

## 9 DISCUSSION AND CONCLUSIONS

In this work we studied the influence of learning the default policy in the KL-regularized RL objective. Specifically we looked at the scenario where we enforce information asymmetry between the default policy and the main one. In the continuous control, we showed empirically that in the case of *sparse-reward* tasks with complex walkers, there is a significant speed-up of learning compared to the baseline. In addition, we found that there was no significant gain in *dense-reward* tasks and/or with simple walkers. Moreover, we demonstrated that significant gains can be achieved in the discrete action spaces. We provided evidence that these gains are mostly due to the information asymmetry between the agent and the default policy. Best results are obtained when the default policy sees only a subset of information, allowing it to learn task-agnostic behaviour. Furthermore, these default polices can be reused to significantly speed-up learning on new tasks.

## 10 ACKNOWLEDGMENTS

The authors would like to thank Abbas Abdolmaleki, Arun Ahuja, Jost Tobias Springenberg, Siqi Liu for their help on experimental side. Furthermore, The authors would like to thank Greg Wayne for useful discussions. Finally, the authors are very grateful to Simon Osindero and Phil Blunsom for their insightful feedback on the paper.

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

## A   KL-REGULARIZED RL AND INFORMATION BOTTLENECK

In this appendix we derive the connection between KL-regularized RL and information bottleneck in detail. For simplicity we assume that $x_t^{\mathcal{D}}$ is empty, consider dependence only on current state $s_t$ and do not use subscript by $t$ in detailed derivations for notational convenience. We also apologize for some notational inconsistencies, and will fix them in a later draft.

### A.1   MINIMIZING INFORMATION FLOW FROM $S_t$ TO $A_t$ FOR UNSTRUCTURED POLICIES

The simple formulation of the information bottleneck corresponds to maximizing reward while minimizing the per-timestep information between actions and state (or a subset of state, like the goal):

$$\mathcal{L} = \mathbb{E}_\pi[\sum_t (r(s_t, a_t) - \mathsf{MI}[A_t; S_t])] \tag{7}$$

Upper-bounding the mutual information term:

$$\mathsf{MI}[A; S] = \int \pi(s)\pi(a|s) \log \frac{\pi(s)\pi(a|s)}{\pi(s)\pi(a)} \tag{8}$$

$$= \int \pi(s)\pi(a|s) \log \frac{\pi(a|s)}{\pi(a)} \tag{9}$$

$$\leqslant \int \pi(s)\pi(a|s) \log \frac{\pi(a|s)}{\pi^0(a)} \tag{10}$$

$$= \mathbb{E}_\pi[\mathsf{KL}[\pi(A|s)\|\pi^0(A)|s]], \tag{11}$$

since

$$0 \leqslant \pi\mathsf{KL}[\pi(a)\|\pi^0(a)] = \mathbb{E}_\pi[\log \frac{\pi(a)}{\pi^0(a)}] = \mathbb{E}_\pi[\log \pi(a)] - \mathbb{E}_\pi[\log \pi^0(a)] \tag{12}$$

$$\Longleftrightarrow \mathbb{E}_\pi[\log \pi(a)] \geqslant \mathbb{E}_\pi[\log \pi^0(a)]. \tag{13}$$

Thus

$$\mathcal{L} = \mathbb{E}_\pi[\sum_t (r(s_t, a_t) - \mathsf{MI}[A_t; S_t])] \tag{14}$$

$$\geqslant \mathbb{E}_\pi[\sum_t (r(s_t, a_t) - \mathsf{KL}[\pi_t\|\pi_t^0|s_t], \tag{15}$$

i.e. the problem turns into one of KL-regularized RL.

### A.2   MINIMIZING INFORMATION FLOW FROM $S_t$ TO $A_t$ FOR POLICIES WITH LATENT VARIABLES

For policies with latent variables such as $\pi(a|s) = \int \pi(a|z)\pi(z|s)dz$ we obtain:

$$\mathsf{MI}[A; S] = \int \pi(a, s) \log \pi(a|s)dads - \int \pi(a) \log \pi(a)da \tag{16}$$

$$\leqslant \pi \int \pi(a, s) \log \pi(a|s)dads - \int \pi(a) \log \bar{\pi}^0(a)da \tag{17}$$

as before.

We choose $\pi^0(a) = \int \pi(a|z)\pi^0(z)dz$, then:

$$\int \pi(a) \log \bar{\pi^0}(a)da = \int \pi(a) \log \int \pi(a|z)\pi^0(z)dzda \tag{18}$$

$$= \int \pi(a,s) \log \int \pi(a|z)\pi^0(z)dzdads \tag{19}$$

$$= \int \pi(a,s) \log \int \pi(a|z)\frac{\pi(z|s,a)}{\pi(z|s,a)}\pi^0(z)dzdads \tag{20}$$

$$\geqslant \int \pi(a,s,z) \log \frac{\pi(a|z)\pi^0(z)}{\pi(z|s,a)}dzdads \tag{21}$$

$$= \int \pi(a,s,z) \log \frac{\pi(a|z)\pi^0(z)\pi(a|s)}{\pi(a|z)\pi(z|s)}dzdads \tag{22}$$

$$= \int \pi(a,s) \log \pi(a|s)dads + \int \pi(z,s) \log \frac{\pi^0(z)}{\pi(z|s)}dzds, \tag{23}$$

and thus

$$\mathsf{MI}[A;S] \leqslant \int \pi(a,s) \log q(a|s)dads - \int \pi(a) \log \bar{\pi^0}(a)da \tag{24}$$

$$\leqslant \int \pi(a,s) \log \pi(a|s)dads - \int \pi(a,s) \log \pi(a|s)dads - \int \pi(z,s) \log \frac{\pi^0(z)}{\pi(z|s)}dz \tag{25}$$

$$= \int \pi(z,s) \log \frac{\pi(z|s)}{\pi^0(z)}dz = \mathbb{E}_\pi[\mathsf{KL}[\pi(Z|s)\|\pi^0(Z)|s]]. \tag{26}$$

Therefore:

$$\mathcal{L} = \mathbb{E}_\pi[\sum_t (r(s_t,a_t) - \mathsf{MI}[A_t;S_t])] \tag{27}$$

$$\geqslant \mathbb{E}_\pi[\sum_t (r(s_t,a_t) - \mathsf{KL}[\pi(Z_t|s)\|\pi^0(Z_t)|s_t], \tag{28}$$

Thus, the KL regularized objective discussed above can be seen as implementing an information bottleneck. Different forms of the default policy correspond to restricting the information flow between different components of the interaction history (past states or observations), and to different approximations to the resulting mutual information penalties.

This perspective suggests two different interpretations of the KL regularized objective discussed above: We can see the role of the default policy implementing a way of restricting information flow between (past) states and (future) actions. An alternative view, more consistent with the analogy between RL and probabilistic modeling invoked above is that of learning a "default" behavior that is independent of some aspect of the state. (Although the information theoretic view has recently gained more hold in the probabilistic modeling literature, too (e.g. Alemi et al., 2016; 2017)).

## B    DISTRIBUTED LEARNING SETUP

We use a distributed off-policy setup similar to Riedmiller et al. (2018a). There is one learner and multiple actors. These are essentially the instantiations of the main agent used for different purposes. Each actor is the main agent version which receives the copy of parameters from the learner and unrolls the trajectories in the environment, saving it to the replay buffer of fixed size $1e6$. The learner is the agent version which samples a batch of short trajectories windows (window size is defined by *unroll length*) from the replay buffer, calculates the gradients and updates the parameters. The updated parameters are then communicated to the actors. Such a setup speeds-up learning significantly and makes the final performance of the policy better. We compare the performance of on *go to moving target task* with 1 and 32 actors. From figure 7, we see that the effect of the default policy does not disappear when the number of actor decreases to 1, but the learning becomes much slower, noisier and weaker.

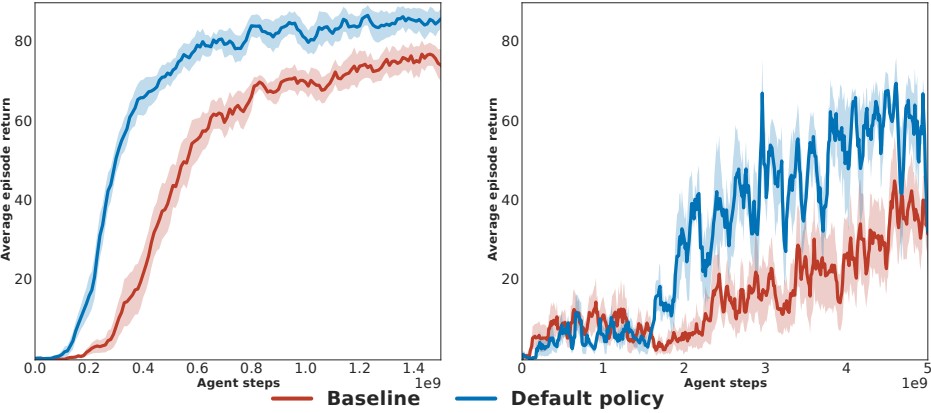

Figure 7: **Single versus multiple actors** comparison on go to moving target task. **Left**: 32 actors. **Right**: 1 actor.

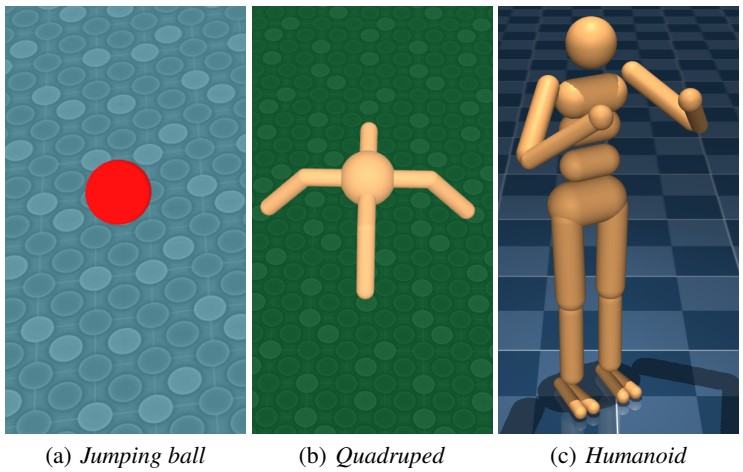

(a) *Jumping ball*  (b) *Quadruped*  (c) *Humanoid*

Figure 8: **Walkers visualization**.

## C CONTINUOUS CONTROL: WALKERS AND TASK DETAILS

Walkers visualization is provided in figure 8. Below we give a detaatiled description of each continuous control task we studied.

**Walking task**.
*Type. Dense-reward feature-based-task.*
*Description.* Each half of the episode, a random direction among 4 (left, right, forward and backwards) is sampled. Task information is specified via a one-hot encoding of the required direction. The walker is required to move in this direction with the target speed $v_t$ and receives the reward $r$.
*Reward.* $r = \exp^{-|v_{cur} - v_t|^2}$.
*Technical details.* Target speed, $v_t = 3$. The episode length is 10 seconds. For the humanoid task we use the absolute head height termination criteria: $h < 0.95$.

**Walls**.
*Type. Dense-reward vision-task.*
*Description.* Walker is required to run through a terrain and avoid the walls. The task-specific information is a vision input. It receives the reward $r$ defined as a difference between the current walker speed $v_{cur}$ and the target speed $v_t$ along the direction of the track.
*Reward.* $r = \exp^{-|v_{cur} - v_t|^2}$.

*Technical details*. Target speed, $v_t = 3$. The episode length is 45 seconds. For the humanoid task we use the absolute head height termination criteria: $h < 0.9$.

**Go to one of K single targets**.
*Type*. *Sparse-reward feature-based-task*.
*Description*. On an infinite floor, there is a finite area of size 8x8 with K randomly placed targets. The walker is also randomly placed in a finite area. The walker's initial position is also randomly placed on the finite area. The walker is required to one of the K targets, specified via command vector. Once it achieves the target, the episode terminates and the walker receives the reward $r$.
*Reward*. $r = 60$.
*Technical details*. The episode length is 20 seconds.

**Go to one moving target**.
*Type*. *Sparse-reward feature-based-task*.
*Description*. Similar to the previous one, but there is only one target and once the walker achieves it, the target reappears in a new random place. The walker receives $r$ for 10 consecutive steps staying on the target before the target reappears in a new random position.
*Reward*. $r = 1$.
*Technical details*. The episode length is 25 seconds.

**Move one box to one of the K targets**.
*Type*. *Sparse-reward feature-based-task*.
*Description*. There is a finite floor of size 3x3 padded with walls with K randomly placed targets and one box. The walker is required to move this box to one of the specified targets. Once the box is placed on the target, the episode terminates and the walker receives the reward $r$.
*Reward*. $r = 60$.
*Technical details*. The episode length is 30 seconds. Control timestep is 0.05 for *quadruped* and 0.025 for *jumping ball*.

**Move one box to one of the K targets and go to another**.
*Type*. *Sparse-reward feature-based-task*.
*Description*. Similar to the previous one, but the walker is also required to go to another target (which is different from the one where it must place the box on). The walker receives the a $r_{task}$ for each task solved, and a $r_{end}$ if it solves both tasks. The other parameters are the same.
*Reward*. $r_{task} = 10$, $r_{end} = 50$.
*Technical details*. Same as in the previous task.

**Foraging in the maze**.
*Type*. *Sparse-reward vision-task*.
*Description*. There is a maze with 8 apples which walker must collect. For each apple, it receives reward $r$. The episode terminates once the walker collects all the apples or the time is elapsed.
*Reward*. $r = 1$.
*Technical details*. The episode length is 90 seconds. Control timestep is 0.025 for *jumping ball*, and 0.05 for *quadruped*.

## D  ALGORITHMS, BASELINE AND HYPERPARAMETERS

Our agents run in off-policy regime sampling the trajectories from the replay buffer. In practice, it means that the trajectories are coming from the behavior (replay buffer) policy $\pi_b$, and thus, the correction must be applied (specified below). Below we provide architecture details, baselines, hyperparmaeters as well as algorithm details for discrete and continuous control cases.

### D.1  DISCRETE CASE

In discrete experiments, we use V-trace off-policy correction as in Espeholt et al. (2018). We reuse all the hyperparameters for DMLab from the mentionned paper. At the top of that, we add default policy network and optimize the corresponding $\alpha$ parameter using population-base training. The difference with the setup in Espeholt et al. (2018) is that they use the human prior over actions (table D.2 in the mentionned paper), which results in 9-dimensional action space. In our work, we take the rough DMLab action space, consisting of all possible rotations, and moving forward/backward,

and "fire" actions. It results in the action space of dimension 648. It make the learning much more challenging, as it has to explore in much larger space.

## D.2 CONTINUOUS CASE

The agent network (see figure 1) is divided into actor and critic networks without any parameter sharing. In the case of *feature-based-task*, the task-specific information is encoded by one layer MLP with ELU activations. For the *vision-task*, we use a 3-layer ResNet He et al. (2015). The encoded task information is then concatenated with the proprioceptive information and passed to the agent network. The actor network encodes a Gaussian policy, $\mathcal{N}(\tilde{\mu}, \tilde{\sigma})$, by employing a two-layer MLP, with mean $\mu$ and log variance $\log \sigma$ as outputs and applying the following processing procedures:

$$\tilde{\mu} = tanh(\mu),$$

$$\tilde{\sigma} = 0.1 + (\sigma_{max} - 0.1)f(\log \sigma),$$

where $f$ is a sigmoid function:

$$f(x) = \frac{1}{1 + \exp^{-x}}.$$

The critic network is a two-layer MLP and a linear readout. The default policy network has the same structure as actor network, but receives a concatenation of the proprioceptive information with only a subset (potentially, empty) of a task-specific information. There is no parameter sharing between the agent and the default policy. ELU is used as activation everywhere. The exact actor, critic and default policy network architectures are described below. We tried to use LSTM for the default policy network instead of MLP, but did not see a difference. We use separate optimizers and learning rates $\beta_\pi, \beta_Q, \beta_{\pi^0}$ for the actor, critic and default policy networks correspondingly. For each network (which we call *online*), we also define the *target* network, similar to the target $Q$-networks (Mnih et al., 2015). The target networks are updated are updated in a slower rate than the online ones by copying their parameters.

We assume that the trajectories are coming from the replay buffer $\mathcal{B}$. To correct for being off-policy, we make use of the Retrace operator (see Munos et al. (2016)). This operator is applied to the $Q$ function essentially introducing the importance weights. We will note $\mathcal{R}Q$ the action for this operator. Algorithm 2 is an off-policy version with retraced $Q$ function of the initial algorithm 1.

We use the same update period for actor and critic networks, $P_a$ and a different period for the default network $P_d$. The baseline is the agent network (see figure 1) without the default policy with an entropy bonus $\lambda$. All the hyperparameters of the baseline are tuned for each task. For each best baseline hyperparameters configuration, we tune the default policy parameters. When we use the default policy, we do not have the entropy bonus. Instead, we have a *regularisation parameter* $\alpha$. The other parameteres which we consider are: *batch size*, *unroll length*. Below we provide the hyperparameters for each of the task. The following default hyperparameters are used unless some particular one is specified.

**Default hyperparameters**.
*Actor learning rate*, $\beta_\pi = 0.0005$.
*Critic learning rate*, $\beta_Q = 0.0005$.
*Default policy learning rate*, $\beta_{\pi^0} = 0.0005$.
*Agent target network update period*: $P_a = 100$.
*Default policy target network update period*: $P_d = 100$.
*Actor network*: MLP with sizes $(300, 200)$.
*Critic network*: MLP with sizes $(400, 300)$.
*Default policy network*: MLP with sizes $(300, 200)$.
*Command encoder network*: 1-layer MLP of size 50.
*Image encoder*: ResNet with filter sizes $(16, 32, 32)$.
*Gaussian policy maximum noise*: $\sigma_{max} = 1.0$.
*Batch size*: 512.
*Unroll length*: 10.
*Entropy bonus*: $\lambda = 0.0001$.
*Regularization constant*: $\alpha = 0.01$.
*Number of actors*: 128.

**Algorithm 2** Off-policy corrected version of algorithm 1 for continuous control

online policy: $\pi_{O,\theta_O}$, initial parameters $\theta_O$
target policy: $\pi_{T,\theta_T}$, initial parameters $\theta_T$
online default policy: $\pi^0_{O,\phi_O}$; initial parameters $\phi_O$
target default policy: $\pi^0_{T,\phi_T}$; initial parameters $\phi_T$
online Q-function: $Q_{O,\psi_O}$; initial parameters $\psi_O$
target Q-function: $Q_{T,\psi_T}$; initial parameters $\psi_T$
target update period: $P$
replay buffer: $\mathcal{B}$
unroll length: $K$
**for** j=1, ... **do**
    Sample partial trajectory from replay buffer $\mathcal{B}$: $\tau_{t:t+K} = (s_t, a_t, r_t \ldots r_{t+K})$
    compute online KL: $\widehat{\mathsf{KL}}_{O,t'} = \mathsf{KL}[\pi_O(\cdot|s_{t'})\|\pi^0_O(\cdot|s_{t'})]$
    compute target KL: $\widehat{\mathsf{KL}}_{T,t'} = \mathsf{KL}[\pi_O(\cdot|s_{t'})\|\pi^0_T(\cdot|s_{t'})]$
    Estimate boostrap value: $\hat{V} = \mathbb{E}_{\pi_T(\cdot|s_{t+K})}[Q_T(s_{t+K}, a)] - \alpha\widehat{\mathsf{KL}}_{T,t+K}$
    Estimate Q targets: $\hat{Q}_{t'} = r_{t'} + \hat{V}$
    Apply Retrace operator: $\hat{Q}^R_{t'} = \mathcal{R}\hat{Q}_{t'}$
    Agent policy loss: $\hat{L}_\pi = \sum_{t'=t}^{t+K-1} \mathbb{E}_{\pi_O(\cdot|s_{t'})}[Q_T(s_{t'}, a)] - \alpha\widehat{KL}_{T,t'}$
    Q-value loss: $\hat{L}_Q = \sum_{t'=t}^{t+K-1} \|\hat{Q}^R_{t'} - Q_O(s_{t'}, a_{t'})\|^2$
    Default policy loss: $\hat{L}_{\pi^0} = \sum_{t'=t}^{t+K-1} \widehat{\mathsf{KL}}_{O,t'}$
    $\theta_O \leftarrow \theta_O + \beta_\pi \nabla_\theta \hat{L}_\pi$
    $\phi_O \leftarrow \phi_O + \beta_{\pi^0} \nabla_\phi \hat{L}_{\pi^0}$
    $\psi_O \leftarrow \psi_O - \beta_Q \nabla_\psi \hat{L}_Q$
    **if** $j \mod P = 0$ **then**
        $\theta_T \leftarrow \theta_O$
        $\phi_T \leftarrow \phi_O$
        $\psi_T \leftarrow \psi_O$
    **end if**
**end for**

**Walking** *quadruped*
*Actor network*: MLP with sizes $(400, 300, 200)$.
*Critic network*: MLP with sizes $(400, 400, 300)$.
*default policy network*: MLP with sizes $(400, 300, 200)$.
*Regularization constant*: $\alpha = 0.0001$.
*Number of actors*: 256.

**Walking** *humanoid*
*Entropy bonus*: $\lambda = 0.005$.
*Regularization constant*: $\alpha = 0.0001$.
The rest is similar to **Walking** *quadruped*.

**Walls** *quadruped*
*Actor and critic learning rate*: $\beta_\pi, \beta_Q = 5e - 5$.
*Batch size*: 48.
*Regularization constant*: $\alpha = 0.001$.
*Number of actors*: 64.

**Walls** *humanoid*
*Actor and critic learning rate*: $\beta_\pi, \beta_Q = 0.0001$.
*Batch size*: 48.
*Regularization constant*: $\alpha = 0.001$.
*Number of actors*: 64.

**Go to moving target** *quadruped*
*Regularization constant*: $\alpha = 0.006$.
*Number of actors*: 32.

**Go to moving target** *humanoid*
*Regularization constant*: $\alpha = 0.1$.

**Go to K targets** *quadruped*
*Actor and critic learning rate*: $\beta_\pi, \beta_Q = 0.0001$.
*default policy target network update period*: $P_d = 50$.
*Regularization constant*: $\alpha = 0.006$.

**Move 1 box to 1 target** *jumping ball*
Default

**Move 1 box to 1 target** *quadruped*
*Actor and critic learning rate*: $\beta_\pi, \beta_Q = 0.0001$.

**Move 1 box to one of 2 targets** *quadruped*
*Actor and critic learning rate*: $\beta_\pi, \beta_Q = 0.0001$.
*default policy target network update period*: $P_d = 50$.

**Move 1 box to one of 2 targets with go to another one** *quadruped*
Same as previous task.

**Move 1 box to one of 3 targets** *quadruped*
Same as previous task.

**Foraging** *jumping ball*
*Actor and critic learning rate*: $\beta_\pi, \beta_Q = 0.0001$.
*Actor network*: LSTM with one hidden unit of size 128.
*Critic network*: LSTM with one hidden unit of size 128.
*Batch size*: 48.
*Number of actors*: 64.

**Foraging** *quadruped*
*Unroll length*: 20
*Regularization constant*: $\alpha = 0.006$.

For the **Foraging** *quadruped* task, the initial agent did not learn, so we used a slightly different version of the agent. In algorithm 2, we essentially learn a $Q$ function and update the policy by

sampling actions from it and backpropagating through $Q$. In this algorithm, we learn a value function $V$ using V-trace (Espeholt et al., 2018) and the policy is updated using an off-policy corrected policy gradient with empirical returns.

# E    CONTINUOUS CONTROL: ADDITIONAL RESULTS

## E.1    DENSE REWARD TASKS

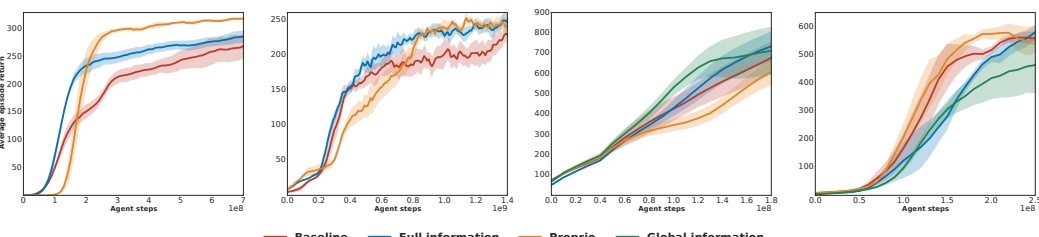

Figure 9: **Results for the dense-reward tasks**. Starting from left. **First**: walking *quadruped* task. **Second**: walking *humanoid* task. **Third**: walls *quadruped* task. **Forth**: walls *humanoid* task. The legends denote additional to the proprioception, information passed to the default policy (except baseline, where we do not use default policy).

The results for the dense-reward tasks are given in figure 9. We observe little difference of using default policy comparing to the baseline. In the walls task, we also consider the default policy with global information, such as the orientation, position and speed in the global coordinates. We do not observe a significant difference between using the default policy and the baseline. The reason for this, we believe, is that the agent is being trained very quickly by seeing a strong reward signal, so the default policy cannot catch it up.

## E.2    SPARSE REWARD JUMPING BALL

The results for the sparse reward tasks with *jumping ball* are given in figure 9. We see little difference of using default policy comparing to the baseline. Our hypothesis consists in the fact that since the default policy affects the policy by regularizing the state-conditional action distribution (policy), for too simple actions space such is given here (3 actions), this effect is not strong enough.

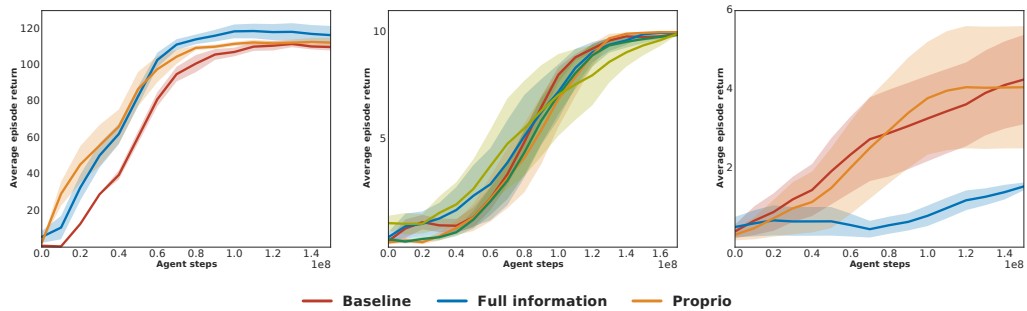

Figure 10: **Results for sparse-reward tasks** with *jumping ball* walker. **Left**: go to moving target. **Center**: moving one box to one target. **Right**: foraging in the maze. The legends denote additional to the proprioception, information passed to the default policy (except baseline, where we do not use default policy).

## E.3    SPARSE REWARD WITH QUADRUPED ADDITIONAL RESULTS

In this section, we provide more results for the sparse reward tasks. In figure 11 the results for going to one of K targets task with *quadruped* are presented. The proprioceptive default policy

gives significant gains comparing to others. What interesting is that when the number of targets $K$ increases, the baseline performance drops dramatically, whereas the proprioceptive default policy solve the task reliably. Our hypothesis is that the default policy learns quickly the default walking behavior which becomes very helpful for the agent to explore the floor and search for the target.

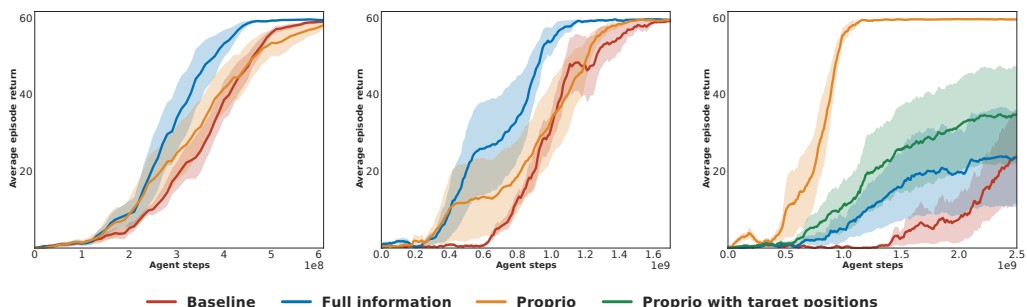

Figure 11: **Results for go to one of K targets tasks** with *quadruped*. **Left**: go to 1 target. **Center**: go to one of 2 targets. **Right**: go to one of 3 targets. The legends denote additional to the proprioception, information passed to the default policy (except baseline, where we do not use default policy).

We also provide the results for move box to one of K targets task, where $K = 1, 2, 3$, and move box to one of two targets task with go to another. The results are given in figure 12. Similar effect occurs here.

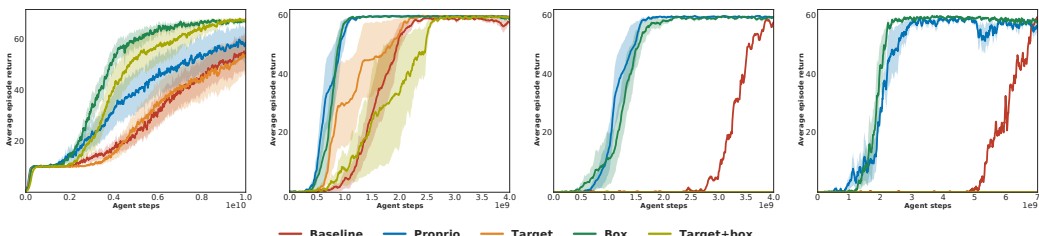

Figure 12: **Results for box pushing tasks** with *quadruped*. Starting from left, **first**: move one box to one of 2 targets with go to another. **Second**: move one box to 1 target. **Third**: move one box to one of 2 targets. **Forth**: move one box to one of 3 targets. The legends denote additional to the proprioception, information passed to the default policy (except baseline, where we do not use default policy).

### E.4 ADDITIONAL TRANSFER RESULTS

In this section, we provide additional transfer experiment results for the range of the tasks. They are given in figure 13. In the first two cases we see that proprioceptive default policy from *the go to target task* gives a significant boost to the performance comparing to the learning from scratch. We also observe, that for the *box pushing tasks*, the default policy with the box position significantly speeds up learning comparing to other cases. We believe it happens because this default policy learns the best default behavior for these tasks possible: going to the box and push it. For the most complicated task, *move one box to one of two targets and go to another one*, 13, right, the box default policy makes a big difference: it makes the policy avoid being stuck in go to target behavior (line with reward of 10).

Additional results for the transfer experiments are given in figure 13. We observe the same effect happening: whereas the baseline performance drops significantly, the agent with default policy stays

### E.5 ABLATION WALLS QUADRUPED

Ablations for the walls *quadruped* are given in figure 14.

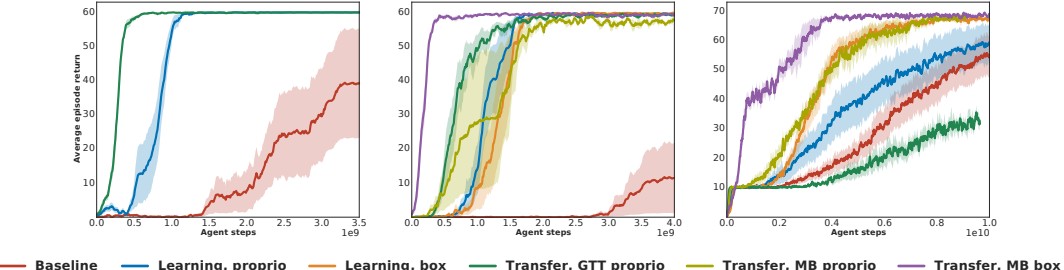

Figure 13: **Performance of the transfer** with *quadruped* walker. **Left**: Go to one of 3 targets. **Center**: move one box to one of two targets. **Right**: move one box to one of two targets and go to another one. The legend whether the default policy is learned or is transferred. Furthermore, it specifies the task from which the default policy is transferred as well as additional information other than the proprioceptive information that the default policy is conditioned on, if any.

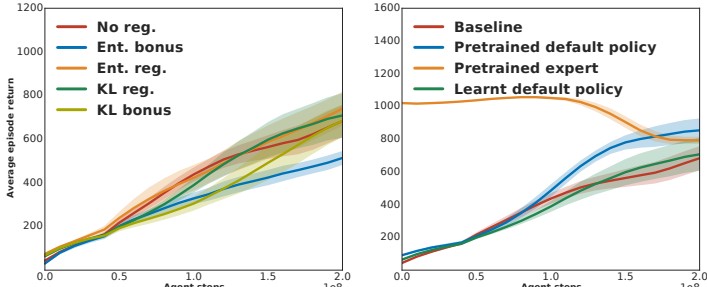

Figure 14: **Ablations** for walls task with *quadruped*. **Left**: Comparing various regularization schemes. **Right**: Optimistic baselines comparing pretrained default policies.

### E.6 ORDER OF THE DEFAULT POLICY IN THE KL-TERM

The results for having the different order of the default policy in the KL-term ($KL[\pi||\pi^0]$ or $KL[\pi^0||\pi]$) for go to moving target task with *quadruped* walker are shown in figure 15. We use this term either in per time step actor loss (auxiliary loss) or as a regularizer by optimizing the objective 1 (with different order of KL). We do not observe significant difference.

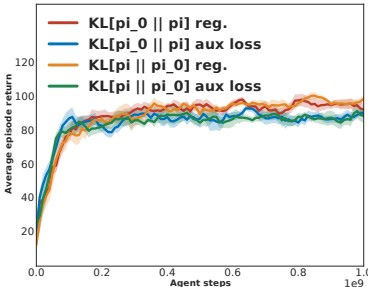

Figure 15: **KL direction** results for go to moving target task with *quadruped*.

