# OpenReview forum: "Information asymmetry in KL-regularized RL"
_ICLR.cc/2019/Conference_

### Official Review · AnonReviewer3 · 2018-11-01
**Good work in general**

**Rating:** 7
**Confidence:** 4

**Review:**


-- Originality --

This paper studies how to use KL-regularization with information asymmetry to speed up and improve reinforcement learning (RL). Compared with existing work, the major novelty in the proposed algorithm is that it uses a default policy learned from data, rather than a fixed default policy. Moreover, the proposed algorithm also limits the amount of information the default policy receives, i.e., there is an "information asymmetry" between the agent policy and the default policy. In many applications, the default policy is purposely chosen to be "goal agnostic" and hence conducts the "transfer learning". To the best of my knowledge, this "informationally asymmetric" KL-regularization approach is novel.

-- Clarify --

The paper is well written in general and is easy to follow.

-- Significance --

I think the idea of regularizing RL via an informationally asymmetric default policy is interesting. It might be an efficient way to do transfer learning (generalization) in some RL applications. This paper has also done extensive and rigorous experiments. Some experiment results are thought-provoking.

-- Pros and Cons

Pros:

1)  The idea of regularizing RL via an informationally asymmetric default policy is interesting. To the best of my knowledge, this "informationally asymmetric" KL-regularization approach is novel.

2) The experiment results are extensive, rigorous, and thought-provoking.

Cons:

1) My understanding is that this "informationally asymmetric" KL-regularization approach is a general approach and can be combined with many policy learning algorithms. It is not completely clear to me why the authors choose to combine it with an actor-critic approach (see Algorithm 1)? Why not combine it with other policy learning algorithms? Please explain.

2) This paper does not have any theoretical results. I fully understand that it is highly non-trivial or even impossible to analyze the proposed algorithm in the general case. However, I recommend the authors to analyze (possibly a variant of) the proposed algorithm in a simplified setting (e.g. the network has only one layer, or even is linear) to further strengthen the results.

3) The experiment results of this paper are interesting, but I think the authors can do a better job of intuitively explaining the experiment results. For instance, the experiment results show that when the reward is "dense shaping", the proposed method and the baseline perform similarly. Might the authors provide an intuitive explanation for this observation? I recommend the authors to try to provide intuitive explanation for all such interesting observations in the paper.

---

> ### Author Response · Authors · 2018-11-20
> **Thank you for the insightful comments. 1. Many modern algorithms use actor-critic architecture though we will investigate other algorithms in the future work. 2. Theory in the follow-up. 3. Improved clarity**
>
> We appreciate the reviewers positive feedback and the insightful comments. Thank you. Below we provide replies to the three concerns raised by the reviewer.
>
> Comment: My understanding is that this "informationally asymmetric" KL-regularization approach is a general approach and can be combined with many policy learning algorithms. It is not completely clear to me why the authors choose to combine it with an actor-critic approach (see Algorithm 1)? Why not combine it with other policy learning algorithms? Please explain
>
> Answer: This is an important point. The core idea of the paper is indeed largely algorithm agnostic; in principle, any algorithm suitable for optimizing the KL-regularized expected reward objective could be considered. For the experiments we chose well established algorithms for learning. As it turns out, many modern algorithms use an actor-critic architecture of some kind. In the paper, we already perform experiments with three reference algorithms:  IMPALA [1], SVG(0) [2] + Retrace [3], continuous version of IMPALA (which we refer as Vtrace) [1] (for vision foraging task, details are in the appendix D).  These differ along important dimensions (e.g. learning Q-function vs. learning V-function only; score-function estimator vs. reparametrization for policy gradient step; use of replay buffer or not). Of course testing other algorithms will further reinforce the significance of our approach, and we intend to do so in future work. But this also represents a non-trivial amount of work, and might be out of scope for a single publication.
>
> Comment: This paper does not have any theoretical results. [...] I recommend the authors to analyze (possibly a variant of) the proposed algorithm in a simplified setting (e.g. the network has only one layer, or even is linear) to further strengthen the results.
> Answer: Thank you for your suggestion. In the paper we established a theoretical insight by connecting it to the information bottleneck and variational EM. One additional simple analysis we could add is similar to the one from Soft Actor-Critic [4] paper where we could show that soft policy iteration algorithm with additional default policy optimization step will converge to the optimal policy (within the restricted set of policies). We are currently thinking about more detailed theoretical analysis, and decided to leave this for the future work (putting all analysis together) since it represents a non trivial amount of effort.
>
> Comment: The experiment results of this paper are interesting [...] I recommend the authors to try to provide intuitive explanation for all such interesting observations in the paper.
> Answer: We did attempt to provide an intuitive explanation for such observations, e.g. on page 7, where we discuss “dense-reward” case. (There is also appendix E, with additional experimental results and explanations). But we agree with the reviewer that the clarity of these points can be improved. Please see the updated version of the paper with improved clarity of the interesting observations in the paper. For the dense reward case in particular please also see our reply to AnonReviewer2 above.
>
>
> References:
>
> [1] Espeholt L., Soyer. H., Munos R., Simonyan K., Mnih V., Ward T., Doron Y., Firoiu V., Harley T., Dunning I., Legg S., Kavukcuoglu K., "IMPALA: Scalable Distributed Deep-RL with Importance Weighted Actor-Learner Architectures", 2018, https://arxiv.org/abs/1802.01561
>
> [2] Heess N., Wayne G., Silver D., Lillicrap T., Tassa Y., Erez T., "Learning Continuous Control Policies by Stochastic Value Gradients", 2015, https://arxiv.org/abs/1510.09142
>
> [3] Munos R., Stepleton T., Harutyunyan A., Bellemare M.G., "Safe and Efficient Off-Policy Reinforcement Learning", 2016, https://arxiv.org/abs/1606.02647
>
> [4] Haarnoja T., Zhou A., Abbeel P., Levine S., "Soft Actor-Critic: Off-Policy Maximum Entropy Deep Reinforcement Learning with a Stochastic Actor", 2018, https://arxiv.org/abs/1801.01290

---

### Official Review · AnonReviewer1 · 2018-11-02

**Rating:** 5
**Confidence:** 5

**Review:**

This is a very interesting piece of work. We know from cognitive science literature, that there are 2 distinct modes of decision making - habit based and top-down control (goal directed) decision making. The paper proposes to use this intuition by using information theoretic objective such that the agent follows "default" policy on average and agent gets penalized for changing its "default" behaviour, and the idea is to minimize this cost on average across states.

The paper is very well written. I think, this paper would have good impact in coming up with new learning algorithms which are inspired from cognitive science literature as well as mathematically grounded. But I dont think, paper in its current form is suitable for publication.

There are several reasons, but most important:

1) Most of the experiments in this paper use of the order of  10^9 or even 10^10 steps. Its practically not possible for anyone in academia to have such a compute. Now, that said, I do think this paper is pretty interesting. Hence, Is it possible to construct a toy problem which has similar characteristics, and then show similar results using like 10^6 or 10^7 steps ? I think it would be easy to construct a 2D POMPD maze navigation env and test similar results. This would improve the paper, as well as could provide a baseline which people in the future can compare to.

2) It becomes more important to compare to stronger baselines like maximum entropy RL ( for ex. Soft Actor Critic). And spend some good of amount time getting these baselines right on these new environments.

---

> ### Author Response · Authors · 2018-11-20
> **Thank you for the comment. 1. Scale of our experiments is comparable to the existing DeepRL approaches. 2. We eventually compare to similar to SAC baseline**
>
> We appreciate the reviewer comments about the habit-based perspective on the decision making which as he points out gives an intuition for using information theoretic objective to encourage an agent to follow default behaviors on average and being penalized by deviating. We thank the reviewer for the insights and for the positive feedback. Below we address two raised concerns.
>
> Comment:
> Most of the experiments in this paper use of the order of 10^9 or even 10^10 steps[...]
>
> Answer:
> We apologize for the confusion. In our setting, actors are collecting experience that is sent to a centralized learner that computes gradients using the stored trajectories. The learner will go over the stored trajectories multiple times. We report performance as a function of the number of non-unique environment transitions processed by the learner. To compute, for instance, the actual number of weight updates we thus need to divide the number on the x-axis of our plots  by the batch size (512) and sequence length (10). The number of updates applied to the model in our setup is thus approximately 3 orders of magnitude lower than suggested by the plots and well within the standard number of updates used in most deep learning or deep RL papers. (e.g. IMPALA [1], Soft Actor-Critic [2], etc.). We will further improve the explanation of the setup in the paper [in the originally submission located at the end of page 6 - beginning of page 7, "Experimental setup" paragraph, last 5 sentences] in a revised version of the paper.
> To make this point more concrete, in Soft Actor-Critic paper [2], the authors report the number of executed environment steps, where for each step, they apply a batched (of size 256) gradient update (in our terminology it corresponds to the 256 agent steps) on the trajectories sampled from the replay buffer to modify agent parameters (see open-sourced code at https://github.com/haarnoja/sac/blob/master/sac/algos/base.py, lines 89-110; Page 5, last sentence at the right bottom: “[...] In practice, we take a single environment step followed by one or several gradient steps). If we take an example of Ant-v1 task, and try to map their results onto a similar plotting schema as in our paper, we should multiple their number of steps (3e6) by their batch size (256), and we obtain 7.68 * 1e8 “agent steps” (non-unique transitions processed by learner). The order of magnitude is similar to some of our simpler tasks, such as the task at Figure 5. This is, of course, not an entirely direct comparison because we have a distributed actor-learner setup, but it demonstrates that the order of magnitude of learning updates in our setup is consistent with the methods reported in Deep RL literature. Nevertheless, we do understand the reproducibility point of view, and while we do not agree this paper has an issue with running experiments that are at too large scales, it is an important consideration for future works for the field in general. Finally, we provide a large number of different experiments (including ones in Appendix E) demonstrating our idea. The submission also already contains some results for a non-distributed (Appendix B, figure 7) setup.
>
> Comment:
> It becomes more important to compare to stronger baselines like maximum entropy RL (Soft Actor Critic)[...]
>
> Answer:
> We effectively provide this baseline already [Figure 5, left.]. We compare to two entropy regularized baselines:  SVG(0) [3] with entropy regularization, and SVG(0) with entropy bonus. The former optimizes the entropy regularized expected reward objective using a Q-function and policy updates via reparametrization in the same way as SAC. The latter also implements SVG(0) policy updates (Q function + reparametrization and back-propagation) but includes entropy only in the policy update as e.g. is general practice in many DRL papers (e.g. [4]). We optimize hyperparameters for each algorithm separately. Compared to the SAC algorithm there are some minor differences in the use of target networks, which are, however, orthogonal to the ideas of our paper (which in unreported  experiments made no qualitative difference to the results).
>
> References:
>
> [1] Espeholt L., Soyer. H., Munos R., Simonyan K., Mnih V., Ward T., Doron Y., Firoiu V., Harley T., Dunning I., Legg S., Kavukcuoglu K., "IMPALA: Scalable Distributed Deep-RL with Importance Weighted Actor-Learner Architectures", 2018, https://arxiv.org/abs/1802.01561
> [2] Haarnoja T., Zhou A., Abbeel P., Levine S., "Soft Actor-Critic: Off-Policy Maximum Entropy Deep Reinforcement Learning with a Stochastic Actor", 2018, https://arxiv.org/abs/1801.01290
> [3] Heess N., Wayne G., Silver D., Lillicrap T., Tassa Y., Erez T., "Learning Continuous Control Policies by Stochastic Value Gradients", 2015, https://arxiv.org/abs/1510.09142
> [4] Asynchronous Methods for Deep Reinforcement Learning, Mnih V.,  Badia A.P., Mirza M., Graves A., Lillicrap T.P., Harley T., Silver D., Kavukcuoglu K., 2016, https://arxiv.org/pdf/1602.01783.pdf

---

### Official Review · AnonReviewer2 · 2018-11-07
**Novel approach**

**Rating:** 7
**Confidence:** 3

**Review:**

This paper shows that significant speed-up gains can be achieved by using KL-regularization with information asymmetry in sparse-reward settings.  Different from previous works, the policy and default policy are learned simultaneously.  Furthermore, it demonstrates that the default policy can be used to perform transfer learning.

Pros:

- Overall the paper is well-written and the organization is easy to follow.  The approach is novel and most relevant works are compared and contrasted.  The intuitions provided nicely complements the concepts and experiments are thorough.

Cons:

- The idea of separating policy and default policy seems similar to having high and low level controller (HLC and LLC) in hierarchical control -- where LLC takes proprioceptive observations as input, and HLC handles task specific goals.  In contrast, one advantage of the proposed method in this work is that the training is end-to-end.  Would have liked to see comparison between the proposed method and hierarchical control.

- As mentioned, the proposed method does not offer significant speed-up in dense-reward settings.  Considering that most of the tasks experimented in the paper can leverage dense shaping to achieve speed-up over sparse rewards, it'd be nice to have experiments to show that for some environments the proposed method can out-perform baseline methods even in dense-reward settings.

---

> ### Author Response · Authors · 2018-11-20
> **Thank you. Summary: 1. HRL differs from our setup to be directly comparable. The connections are left for future work. 2. Dense-reward setting is too simple for the regularizer to give a significant improvement**
>
> We thank the reviewer for their time, positive feedback, and insightful comments. Below are replies to the two questions the reviewer raised:
>
> Comment:
> The idea of separating policy and default policy seems similar to having high and low level controller [...] Would have liked to see comparison between the proposed method and hierarchical control.
>
> Answer:
> This is indeed a very interesting question. The proposed framework currently differs from “classical” HRL ideas in the following sense:
>     - The agent policy is not hierarchical (there is no notion of high-level actions).
>     - The default policy is an external component, and not a part of the agent (such as LLC in HRL). In practice you need the default policy only during training, not at test time.
> These differences suggest that there is no simple 1-1 mapping to HRL frameworks that are focused on HL actions which makes a sensible comparison difficult. Nevertheless our method does, of course, share important intuitions with work on HRL in the sense of learning about reusable behavioral structure. In the paper, we propose a theoretical justification for the current framework from the information bottleneck perspective. In the appendix A.2, we do indeed derive the form of the objective for latent-variable policies, which can be seen as having HL (latent space) and LL (action space). This suggests that these ideas can eventually be unified, and we are currently investigating these connections in detail. These results are, however, beyond the scope of the current work.
>
> Comment:
> As mentioned, the proposed method does not offer significant speed-up in dense-reward settings. [...] it'd be nice to have experiments to show that for some environments the proposed method can out-perform baseline methods even in dense-reward settings.
>
> Answer:
> As we mention in the paper, in the dense-reward setup the problem of learning the policy with KL-regularization to the default one, is not simpler than regular policy learning. We explain it in the appendix E.1 that it is probably due to already strong reward signal. If everywhere in the state space we get a sufficient learning signal to learn the relevant behavior then the point of the default policy (which should help to provide a structured exploration strategy, asking the agent to act consistently to other regions it has seen and learned) is somewhat reduced. Nevertheless, in the Appendix E, we provide additional results for the dense-reward tasks and show that the current method performance doesn’t become worse comparing to the baseline. Our intuition is that an example where our method would also help in the dense-reward scenario would be the one with weak shaping reward combined with complex action space (e.g. humanoid). Finding such scenarios is left for the follow-up work.

---

### Comment · Area_Chair1 · 2018-11-20
**thanks for the reviews; authors response?**

Thanks for the detailed review comments thus far.
Do the authors wish to add anything or respond in any way?
-- area chair

---

### Meta-Review · Area_Chair1 · 2018-12-14
**intuitive idea & theoretical connections; solid experimental results**

**Confidence:** 5
**Recommendation:** Accept (Poster)

**Metareview:**

Strengths

The paper introduces a promising and novel idea, i.e., regularizing RL via an informationally asymmetric default policy
The paper is well written.  It has solid and extensive experimental results.

Weaknesses


There is a lack of benefit on dense-reward problems as a limitation, which the authors further
acknowledge as a limitation. There also some similarities to HRL approaches.
A lack of theoretical results is also suggested. To be fair, the paper makes a number of connections
with various bits of theory, although it perhaps does not directly result in any new theoretical analysis.
A concern of one reviewer is the need for extensive compute, and making comparisons to stronger (maxent) baselines.
The authors provide a convincing reply on these issues.

Points of Contention

While the scores are non-uniform (7,7,5), the most critical review, R1(5), is in fact quite positive on many
aspects of the paper, i.e., "this paper would have good impact in coming up with new
learning algorithms which are inspired from cognitive science literature as well as mathematically grounded."
The specific critiques of R1 were covered in detail by the authors.

Overall

The paper presents a novel and fairly intuitive idea, with very solid experimental results.
While the methods has theoretical results, the results themselves are more experimental than theoretic.
The reviewers are largely enthused about the paper.  The AC recommends acceptance as a poster.